# The Acute Effect of Accentuated Eccentric Overloading upon the Kinematics and Myoelectric Activity in the Eccentric and Concentric Phase of a Traditional Bench Press

**DOI:** 10.3390/sports10010006

**Published:** 2021-12-29

**Authors:** Eirik Lindset Kristiansen, Stian Larsen, Roland van den Tillaar

**Affiliations:** Department of Sports Sciences and Physical Education, Nord University, 7600 Levanger, Norway; ek1105@hotmail.com (E.L.K.); stianandrelarsen@live.no (S.L.)

**Keywords:** weight releasers, maximal strength, augmented overload

## Abstract

The target of this study was to investigate the acute effect of a supramaximal augmented eccentric load on the kinematics and myoelectric activity during the concentric phase of the lift in a traditional bench press. Ten resistance-trained males (age 24 ± 6.4 years, height 1.80 ± 0.07 m, body-mass 87.2 ± 16.9 kg) performed two repetitions at 110/85% of the 1-RM in the dynamic accentuated external resistance (DAER) group and two repetitions at 85/85% of the 1-RM for the control group in a traditional bench press. The barbell kinematics, joint kinematics and myoelectric activity of eight muscles were measured in the eccentric phase and the pre-sticking, sticking and post-sticking regions. The main findings were that the sticking region started at a lower barbell height and that a lower barbell velocity was observed in the sticking region during the second repetition in the DAER condition compared to the control condition. Additionally, the lateral deltoid muscle and clavicle part of the pectoralis were more active during the eccentric loading compared to the control condition for the second repetition. Furthermore, higher myoelectric activity was measured during the second repetition in the sticking region for the eccentric loading condition in both pectoralis muscles, while the sternal parts of the pectoralis and anterior deltoid were more active during the second repetition of the control condition in the post-sticking region. Based on our findings, it can be concluded that the supramaximal loading in the descending phase with 110% of the 1-RM in the bench press does not have an acute and positive effect of enhanced performance in the ascending phase of the lift at 85% of 1-RM. Instead, fatigue occurs when using this eccentric load during a bench press.

## 1. Introduction

The traditional bench press is a classic weight-resistance exercise for the upper body and it is used both in bodybuilding and in the sport of powerlifting as one of the competition exercises. The bench press is a pressing movement that mainly targets the pectoralis major, the anterior head of the deltoid and the triceps brachialis. The traditional bench press is performed while lying on a flat bench with the bar being lowered to the chest and then pressed upwards until the elbows are fully extended [1]. The exercise is often performed as a countermovement with an eccentric and concentric phase, where the barbell load is usually the same in both phases. However, several studies have shown that the eccentric phase influences the concentric when it comes to execution. This phenomenon is known as the countermovement cycle [2,3,4]. Wilson, Elliott and Wood [3] found that the participants achieved a 14% increase in strength with a countermovement bench press compared to a pure concentric bench press. They speculated that the strength increase was due to an increase in the force output in the first 200 ms for the countermovement bench press due to a higher pre-activation of the muscles as well as the contractile element potentiation.

Ojasto and Hakkinen [5] investigated the effects of 105%, 110% and 120% of the one-repetition maximum (1-RM) during the eccentric phase, with the same absolute load of 100% of the 1-RM during the concentric phase. This is a method called dynamic accentuated external resistance (DAER), which applies to loading designs where the eccentric phase of the repetition is followed by the concentric phase with a smaller load. They showed that the extra loads decreased the performance of the concentric 1-RM and peak force. Recently, van den Tillaar and Kwan [6] conducted a study on eccentric loading at 95% in the descending phase of the lift, to see whether it affected the bench press performance in the ascending phase of the lift due to the possible pre-activation of the muscle. They found that the extra loading did not enhance the performance in the bench press, which is consistent with the study by Ojasto and Hakkinen [5]. However, Munger et al. [7] showed, with the same conditions as Ojasto and Hakkinen [5], an increase in the concentric peak velocity and peak power in a front squat experiment. The difference in their research was that they had an absolute concentric load of 90% of the 1-RM and it was a front squat study [7]. Using the research of Munger, Archer, Leyva, Wong, Coburn, Costa and Brown [7] as a base, we wanted to investigate whether this could also be the case for the bench press, but with a slightly lighter concentric load of 85%.

Therefore, the present study aimed to investigate the acute effect that eccentric loading has on the kinematics and myoelectric activity of the bench press and determine whether it has a positive impact on the concentric phase. The hypothesis for this study was that the eccentric overload would fill the contractile elements with more energy and activate them at a higher rate, immediately before the concentric phase begins. The eccentric overload creates energy that can be stored within the contractile elements and be used during the concentric phase. Because of the higher “pre-activation”, we hypothesized that the muscles might be able to create or unleash a higher force at the beginning of the concentric phase, resulting in a higher velocity and decreasing the sticking region compared with the traditional bench press.

## 2. Materials and Methods

To investigate the acute effects of an eccentric overload during the concentric phase of a regular traditional bench press upon kinematics and myoelectric activity, a within-subject repeated measures design was used. The dependent variables barbell kinematics and joint kinematics were collected as direct variables in the events v_ecc_, v_0_, v_max1_, v_min_ and v_max2_, while the dependent variables myoelectric activity and joint moments were collected as means in the pre-sticking, sticking and post-sticking region. Five subjects started the test with an eccentric overload and finished with a comparison test using the same load in both the eccentric and concentric phase of the bench press. The remaining five subjects did the opposite. The subjects were randomly placed into the two conditions.

### 2.1. Subjects

Ten resistance-trained males (age 24 ± 6.4 years, height 1.80 ± 0.7 m, body mass 87.2 ± 16.9 kg) participated in the study. All participants had been training using a bench press on a weekly basis for a minimum of six consecutive months before participating in the study. Participants were not allowed to perform any resistance training or consume alcohol 48 h before testing. A written consent form was signed for each participant before the first familiarization session. The ethics of the study corresponded to the institutional requirements and approval for data security and handling was obtained from the Norwegian Center for Research Data (project nr: 991974) on the 20th of August 2020, following the latest revision of the Declaration of Helsinki.

### 2.2. Procedures

First, the participants performed a pretest, where they worked up to a 1-RM. The warmup was performed with as many repetitions with the barbell as the participants wanted and thereafter followed by a standardized protocol of eight reps at 40% of estimated 1-RM, six reps at 60% of estimated 1-RM, three reps at 70% of estimated 1-RM and two reps at 80% of estimated 1-RM, with a rest interval of two minutes between sets. After the warmup, the subjects proceeded to test their 1-RM [6]. After each lift, the participants were asked whether they thought that they could go any heavier at all. If they were uncertain whether the first lift was their actual 1-RM, the barbell load was adjusted accordingly, adding 2.5–5 kg to the load at a time. Each 1-RM attempt had a minimum of three minutes of rest in between to ensure sufficient recovery between sets. During the warmup, the grip width of each participant was noted. In this way, it was ensured that each participant lifted with the same grip width at each attempt, because previous studies have shown that movement kinematics and kinetics change with grip width [8,9,10].

A familiarization session was performed after the 1-RM test. The familiarization was performed to allow the participants to become familiar with the weight releasers, as they can feel unnatural in a traditional bench press. In this session, they practiced with the weight released at similar intensities to those used during the test day. On the test day, six electromyography (EMG) sensors were placed on the participant. The EMG sensors were placed on the pectoralis major (the clavicular and sternal head), anterior and lateral deltoid, biceps brachialis and triceps long head [11]. Unlike some previous studies, we also included the lateral head of the deltoid and the clavicular head of the pectoralis major in this study. This was done due to our hypothesis that suggested a change in the movement pattern of the barbell during the concentric phase due to the heavy eccentric load which led to increased activity in the clavicular head of the pectoralis major and on the lateral head of the deltoid. Each EMG sensor was placed in the assumed direction of the muscle fibers, as recommended by SENIAM [12]. For the EMG sensor to properly function and establish solid contact with the muscles, the subjects had to shave and clean the areas where the sensors were placed. The test leader then coated the EMG sensors with a small amount of contact gel to enable better contact for the sensors. In addition to the EMG sensors, a 3D motion capture system (Qualisys) with a total of eight cameras was used. Each participant had to wear reflectors, which were placed on the ulna and radius styloid process, on the medial and lateral epicondyle of the humerus, on the acromioclavicular joint (both sides), on the ilium anterior superior (both sides), on the medial and lateral epicondyle of the femur and on the lateral malleolus apex on the fibula. Two reflectors were also placed in the center of the bar, with a 40 cm distance between them, to capture the bar. A linear encoder (ET-Enc-02, Ergotest Technology AS, Langesund, Norway) was placed directly under the bench press platform, with the thread strapped far out on the bar. The equipment was adapted to each participant. This included the bench press platform height and length of the weight releasers.

The test was first randomized by dividing the participants into two groups. Half of the group started the test with the eccentric overload (110% eccentric and 85% concentric) and proceeded with the comparison test with no eccentric overload (85% eccentric and concentric) eight to ten minutes later (Figure 1). The other half of the group started their test with the comparison test. After eight to ten minutes, they performed their second test with the eccentric overload. The participants used the same standardized warmup protocol as in the 1-RM test, with three to four minutes of rest between the last warmup set and the first test. The participants performed one set of two repetitions in each test. During the lifts, the bar touched the chest but was not allowed to bounce. The participants got help from the test leader to lift the bar off the bench press rack. Thereafter, assistants secured the subjects and controlled the weight releasers. The weight releasers were used to create an overload and were placed on the bar before the eccentric phase for each repetition by the assistants [5]. Weight releasers were automatically released from the barbell at v_0_.

### 2.3. Measurements

To measure the lifting time of the barbell and the vertical displacement, a linear encoder (ET-Enc-02, Ergotest Technology AS, Langesund, Norway) was used with a resolution of 0.019 mm and sampling rate of 200 Hz. By using the five-point differential filter with the software (Musclelab version: 10.200.90.5095, Ergotest innovation, Porsgrund Norway), the velocity of the barbell was calculated. The shoulder abduction, shoulder flexion and elbow extension angle, together with barbell displacement and velocity, were recognized at the following positions during the eccentric phase maximal downward velocity (v_ecc_) and during the concentric movement of the bench press at the two repetitions: lowest position of the barbell (v_0_), first maximal barbell velocity (v_max1_), first located lowest barbell velocity (v_min_) and second maximal barbell peak velocity (v_max2_). The EMG recordings and the linear encoder were synchronized by utilizing a Musclelab 6000 system and thereafter analyzed by the musclelab v10.5.67 software (Ergotest Technology AS, Langesund, Norway). The participants used a Soehnle Professional 7830 stand scale to measure body mass. The barbell used was a standard Olympic barbell with a barbell load of 20 kg. The weight releasers were standard and had a mass of 3 kg each.

### 2.4. Statistical Analysis

Data were checked for normality with the Shapiro–Wilks test. To evaluate barbell kinematics (distance, velocity and lifting time) and EMG activity between the two conditions (eccentric load vs. control), a 2 (condition: augmented eccentric loading vs. control) × 2 (repetition 1–2) model analysis of variance (ANOVA) design with repeated measures was used for the events (v_ecc_, v_max1_, v_min_ and v_max2_) and regions (pre-sticking, sticking and post-sticking). If significant differences were found, a Holm–Bonferroni post hoc test was performed. In cases where the sphericity assumption was violated, p-values of the Greenhouse–Geisser adjustments were reported. The level of significance was set at *p* < 0.05. Statistical analysis was performed with SPSS version 27.0 (SPSS Inc., Chicago, IL, USA). Effect size was evaluated with η^2^ (eta squared), where 0.01 < η^2^ < 0.06 constituted a small effect, 0.06 < η ^2^ < 0.14 constituted a medium effect and η^2^ > 0.14 constituted a large effect [13].

## 3. Results

### 3.1. Barbell Kinematics

A significant effect of repetition (F ≥ 6.79, *p* ≤ 0.028, η^2^ ≥ 0.43) was found on the event of v_max1_ (distance, velocity and timing) and v_min_ (distance and velocity), while a significant effect of the lifting condition was found for the total lifting time and the velocity and distance at v_min_ (F ≥ 5.77, *p* ≤ 0.040, η^2^ ≥ 0.39). Furthermore, a significant interaction effect (repetition * condition) was found for the total ascending lifting time, the time of v_min_ and the velocities at v_ecc_ and v_max1_ (F ≥ 5.30, *p* ≤ 0.047, η^2^ ≥ 0.37, Figure 2, Figure 3 and Figure 4). The post hoc tests showed that the velocity at v_max1_ was lower in repetition two compared with one in both conditions and that v_ecc_ and v_min_ for the control condition were higher than for the DAER condition (Figure 1 and Figure 2). Furthermore, the time of occurrence of v_max2_ and the total ascending lifting time in the control condition occurred earlier in repetition one compared with repetition two, while v_min_ occurred earlier during the lift in the eccentric loading condition in repetition two (Figure 2, Figure 3 and Figure 4). This also resulted in the earlier occurrence of v_max2_ and the total ascending lifting time in the control condition in repetition one when compared with the eccentric loading condition. In the second repetition, a lower distance at v_max1_ and velocities at v_max1_ and v_min_ were observed in the eccentric compared with the control condition (Figure 2 and Figure 4).

### 3.2. Joint Kinematics

The shoulder flexion and elbow extension angles decreased from one event to the next (*F* ≥ 3.22, *p* < 0.02, η^2^ ≥ 0.48), while the shoulder abduction angle increased from v_0_ to v_max1_, after which it decreased again from v_min_ to v_max2_ for both repetitions (Figure 5). However, no other significant differences were found for the shoulder abduction, shoulder flexion and elbow extension angles nor any differences between the two bench conditions in the first or second repetition at the different events (F ≤ 1.65 *p* ≥ 0.13, η^2^ ≤ 0.12).

### 3.3. Myoelectric Activity

A significant effect of repetition was found in the pectoralis sternal and deltoid anterior, together with a significant effect of the region for the triceps lateral and lateral deltoid (F ≥ 4.8, *p* ≤ 0.022, η^2^ ≥ 0.35). The post hoc tests showed that the pectoralis sternal and clavicle myoelectric activity significantly increased in the sticking region from repetition one to two for the eccentric loading group, whereas there was increased myoelectric activity in the pectoralis sternal muscle during the repetitions in the post-sticking region in the control group (Figure 6). Moreover, there was increased triceps lateral and deltoid lateral myoelectric activity for the control condition from the pre-sticking to the sticking region. Additionally, a significant effect of the condition was found for the myoelectric activity of the lateral deltoid and pectoralis clavicle (F > 6.2, *p* < 0.034, η^2^ > 0.41). The post hoc comparison revealed that both muscles had greater myoelectric activity with the eccentric loading compared to the control condition (Figure 6).

## 4. Discussion

The purpose of the present study was to evaluate the acute effect of supramaximal eccentric loading upon the kinematics and myoelectric activity during the concentric phase in the bench press exercise. The main findings were that the sticking region started at a lower barbell height with a lower barbell velocity in the repetition for the DAER condition compared to the control condition, which was contradictory to our hypothesis. Additionally, the lateral deltoid muscle and clavicle part of the pectoralis were more active during the eccentric loading compared to the control condition for the second repetition. Furthermore, higher myoelectric activity was measured during the second repetition in the sticking region for the eccentric loading condition in both pectoralis muscles, while the sternal parts of the pectoralis and anterior deltoid were more active during the second repetition of the control condition during the post-sticking region.

Our data showed that supramaximal augmented eccentric loads (>110% of the 1-RM) during the eccentric phase did not change the kinematics during the concentric phase in a positive matter (Figure 1, Figure 2 and Figure 3), since the sticking region started at a lower barbell height. Based on this finding, we suggest that supramaximal eccentric loading does not have a positive effect on performance during the concentric phase of the traditional bench press, which is similar to the findings of Ojasto and Hakkinen [5] and van den Tillaar and Kwan [6]; neither of these groups reported an enhanced performance by using supramaximal loadings during the eccentric phase for the bench press. However, other studies have found extra loading to have an acute enhancing effect on performance during the concentric phase of the traditional bench press, resulting in a greater concentric force [14,15]. However, the main reason for this enhanced performance could be the differences in the test protocol; the research by Sheppard and Young [15] used loads at 40–50% of the 1-RM during the concentric phase of the lift and loads of 10–20 kg extra in the eccentric phase. These low loads had a higher and more acute effect on performance compared to the submaximal and maximal lifts at 85–120% of the 1-RM. Likewise, the research by Doan, Newton, Marsit, Triplett-McBride, Koziris, Fry and Kraemer [14] used a load of 105% in the eccentric phase of the lift and found enhanced performance during the concentric phase. However, the participants had three attempts, where the load was increased for each successful attempt(s) and the real load was reached at 105%; this indicated that additional familiarizations could be more optimal for greater performance with the accentuated eccentric overloading method.

Furthermore, it seems that performance was not enhanced but decreased, as shown by the fact that v_max1_ and v_min_ occurred at a lower barbell height in the second repetition for the DAER condition (Figure 3). Moreover, the eccentric loading condition had a longer total duration on average for the pre-sticking and sticking region for repetition one; however, probably due to the within-subject variability, this was not significant in repetition two. Additionally, the shorter average duration and faster v_ecc_ for the control condition (Figure 3 and Figure 4) could result in the eccentric portion being performed faster to get more elastic energy when undertaking the concentric portion of the lift, which could lead to enhanced performance [16,17]. Merrigan et al. [18] also found that applying 120% of the 1-RM during the eccentric phase in the first repetition (of five repetitions) could result in lower velocities at 50% and 65% of the 1-RM during the concentric phase of each repetition. It is more likely that fatigue occurs when applying these heavy eccentric loads.

In the present study, it was hypothesized that the DAER condition would produce additional pre-activation in the eccentric phase of the lift; thereby, the activity at the start of the concentric phase would be higher. This extra activation seems to occur only for the clavicle pectoralis major and the lateral deltoid during the pre-sticking region in the second repetition of the DAER condition (Figure 6). However, this extra activation at the start of the concentric phase for these muscles did not enhance the performance during the concentric phase of the lift and possibly did not potentiate the contractile elements, as suggested by Walshe, Wilson and Ettema [2]. Therefore, no potentiation occurs, but rather fatigue, as the myoelectric activity was higher (size principle) in the second repetition of both parts of the pectoralis and anterior deltoid during the concentric phase. It was more likely to be fatigue, as it also occurred in the control condition, though later in the lift (post sticking region) in the clavicle pectoralis major and lateral deltoid (Figure 6). This is supported by van den Tillaar and Saeterbakken [19], who found that myoelectric activity in the bench press increased for these muscles due to fatigue.

In addition, our study was based on the research of Munger, Archer, Leyva, Wong, Coburn, Costa and Brown [7] but used a slightly lighter concentric load of 85% and focused on the bench press instead of front squats. It is likely that the exercise made the biggest difference in the results. Munger, Archer, Leyva, Wong, Coburn, Costa and Brown [7] tested overloading in the front squat; because the legs have a larger cross-sectional area than the agonists in the bench press exercise, this may eliminate fatigue quicker. The fact that the muscles used in a bench press are smaller and not stimulated as often may be one of the reasons why they handled the heavy loads more poorly than expected. It appears that the load we used (110% of the 1-RM) in the eccentric phase of this study might have been too heavy to create an acute enhancing effect during the concentric phase with 85% of the 1-RM. Another indication of this is that Ojasto and Hakkinen [5] found an enhancing effect of eccentric loading with 77% of the 1-RM on 50% of the 1-RM during the concentric phase. With these lower loads, it is possible to use the extra myoelectric activation and potentiation at the start of the concentric phase, while the movement velocity is so low that these mechanisms diminish very quickly, and fatigue increases in the bench press. Therefore, we suggest that performing a bench press with an eccentric overload could be beneficial for increasing the power output, but with barbell loads lighter than 85% of the 1-RM.

### Limitations

Finally, the finding of no enhanced performance with supramaximal eccentric loading might be because the weight releasers and heavy eccentric loading felt very unnatural for the participants and therefore affected their performance, either psychologically or physically. This may have demanded more stability due to the weight releasers and could explain the absence of increased performance during the concentric portion of the lift even further. Similarly, only one familiarization test was performed to learn how to adapt to the supramaximal eccentric loading with weight releasers, which may have been inadequate for learning a new eccentric overloading method. Furthermore, only a small sample of subjects was tested, which could have had an influence on the findings. In further studies, additional familiarization tests should be performed to the point where the subject feels comfortable with the weight releasers and the extra loading. Additionally, there should be further practical research to identify the optimal method for the eccentric loading of the bench press or to determine whether it has a positive effect of enhancing performance in the bench press. With research increasingly suggesting the positive benefits of the inclusion of eccentric training in sports and the increasing possibilities and availability of the equipment needed for this training, it is an interesting and relevant topic to discuss.

## 5. Conclusions

Based on our findings, it can be concluded that supramaximal loading in the descending phase with 110% of the 1-RM in the bench press does not have an acute positive effect of enhanced performance in the ascending phase of the lift at 85% of the 1-RM.

## Figures and Tables

**Figure 1 sports-10-00006-f001:**
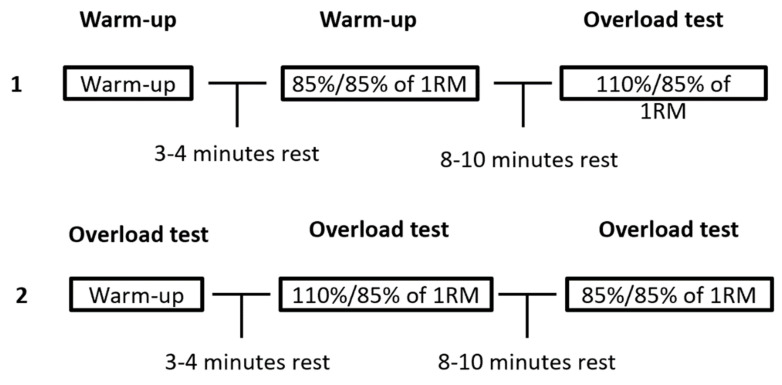
Test protocols: 5 participants used protocol number 1 and the remaining 5 participants used protocol number 2. The participants were randomly placed into each protocol.

**Figure 2 sports-10-00006-f002:**
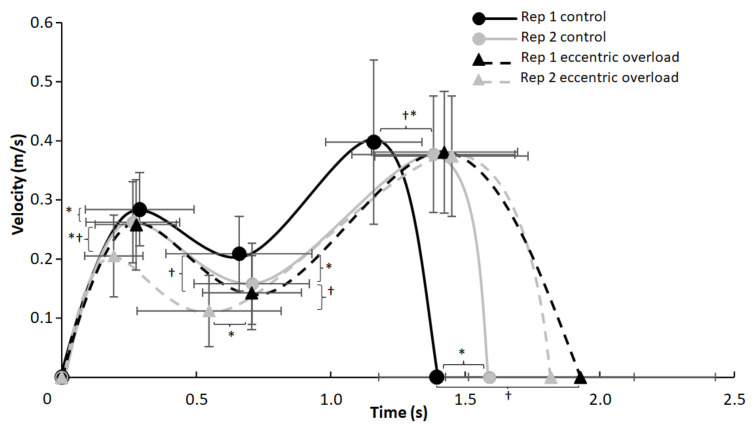
Average (±SD) velocities in each repetition at the different events for each condition during the concentric phase. † Indicates a significant difference between this bench condition and the DAER condition for this repetition at a level of *p* ≤ 0.05. * Indicates a significant difference between these two repetitions for this condition at a level of *p* ≤ 0.05.

**Figure 3 sports-10-00006-f003:**
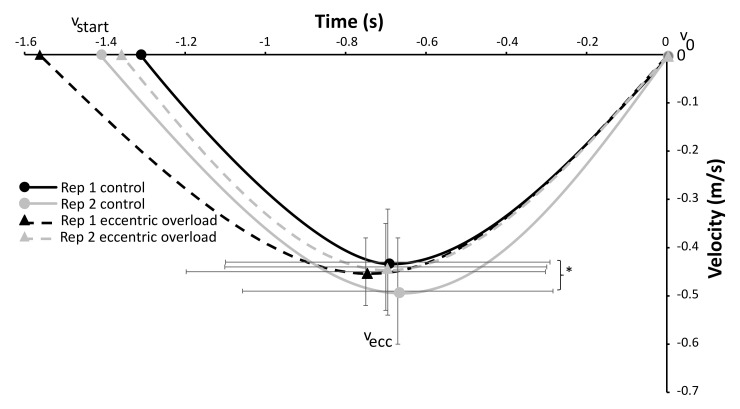
Average (±SD) velocities in each repetition at the different events for each condition during the eccentric phase. * Indicates a significant difference between these two repetitions for this condition at a level of *p* ≤ 0.05.

**Figure 4 sports-10-00006-f004:**
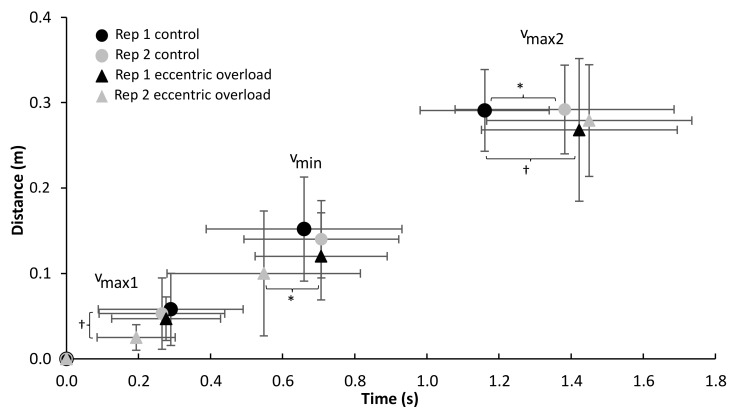
Average (±SD) distance from lowest barbell point in each repetition at the different events for each condition during the concentric phase. † Indicates a significant difference between this bench condition and the DAER condition for this repetition at a level of *p* ≤ 0.05. * Indicates a significant difference between these two repetitions for this condition at a level of *p* ≤ 0.05.

**Figure 5 sports-10-00006-f005:**
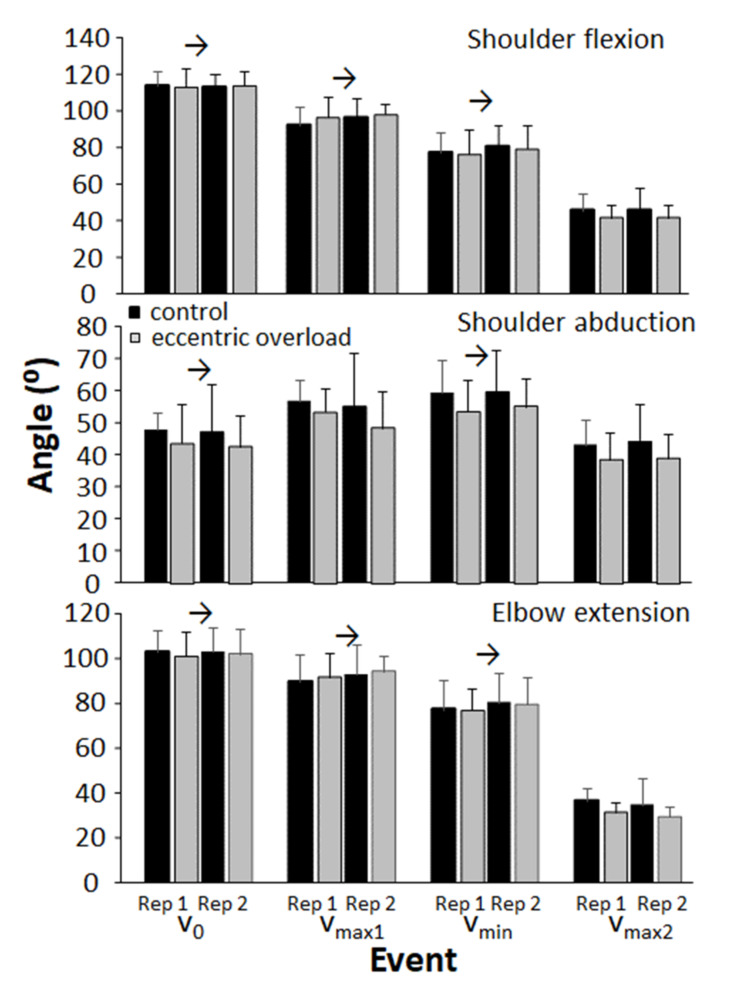
Average (±SD) shoulder abduction, shoulder flexion and elbow extension angle in each repetition at the different events for each condition during the concentric phase. → Indicates a significant difference between this event and following events for both bench conditions at a level of *p* ≤ 0.05.

**Figure 6 sports-10-00006-f006:**
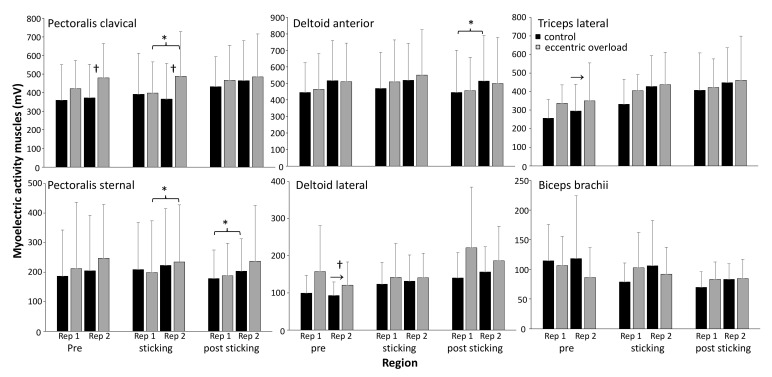
Average (±SD) myoelectric activity for all muscles in each repetition for each condition during the different regions of the concentric phase. † Indicates a significant difference between the control condition and the DAER condition at a level of *p* ≤ 0.05. * Indicates a significant difference between these two repetitions for this condition at a level of *p* ≤ 0.05. → Indicates a significant increase from this region to the next for this condition at a level of *p* ≤ 0.05.

## Data Availability

The data presented in this study are available on request from the corresponding author. The data are not publicly available due to national laws of the Norwegian government on privacy.

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
