# Peer review of "The Acute Effect of Accentuated Eccentric Overloading upon the Kinematics and Myoelectric Activity in the Eccentric and Concentric Phase of a Traditional Bench Press"

_sports, 2021, doi:10.3390/sports10010006_

Round 1

Reviewer 1 Report

line 12: I'd suggest to replace "IN the concentric" with "DURING the concentric", it's more appropriate to stress the duration of a contraction regimen. It's something that happens during a time interval and not at a fixed instanto of time. Do the same throughout the paper.

line 14: I don't like the name you gave to the experimental group "eccentric condition", as it sounds like the control gropu did not performed any eccentric contraction. I'd suggest to find a new name and replace it throughout the paper. You may want to use the terminology used by Brandenburg and Docherty (J Strength Cond Res. 2002 Feb;16(1):25-32) where they use "DCER group" for dynamic constant external
resistance training (i.e., the control group, you can keep control group if you want) and the "DAER group" for dynamic accentuated external resistance.

line 33: "a classic WEIGHT resistance exercise...and IT is used...". There are, indeed, other typer of resistance so better to specify.

line 40: "usually" ??? Of course it is: the external resistance is always the same. This "usually" is either redundant or misleading.

line 47-49: please introduce to the reader how it is possible that a 100% 1RM load lifted during a bench-press exercise (concetric phase) can result in a 105-120% 1RM during the eccentric phase. This because you use the expression "the same absolute load" which may be misleading. I think the authors are giving too many things for granted here. I think you should introduce first the general concept of dynamic accentuated external resistance (DAER).

line 84: 10 partecipants seem like you've got a small sample. Did you performed any sample size analysis (perhaps based on the previously published literature) so assess whether this sample size is sufficient or not? If you can't support this choice from a  statistical point of view, then discuss it as a limitation.

line 94: what protocol did you use for determining the 1RM (direct method)? Pleas specify the relevant literature.

Figure 1: please increase font size of the text within the figure.

Statistical analysis is fine.

Discussion section: I appreciate thee fact that authors have discussed their results in the light of the relevant previously published research, but your study has no limitations! Please add some discussion on what could have been improved in your experimental research so readers may be aware of it when replicating your experiment. A glimps of "future work" seems to be contained at line 324-327, you may add your limitations before this part. Furthermore, I think that, for the sake of study reproducibility and training methodology, it is important to stress the fact that such an experimental methodology can be achieved only using draw-wire econders. This because there is an increasing trend in using inertial sensors (accelerometers) for assessing the barbell kinematics both at a research and at a consumer level. It has been shown, infact, that the reliability of accelerometers decreases approaching 90% of the 1RM. This because the acceleration of the barbell would be close to the gravitational acceleration in case of very slow movements and the accelerometer would not be able to sense any variation of velocity. In other words, the accelerometer won't notice any movement of the barbell when liftin 100-120% 1RM (see https://dx.doi.org/%2010.5812/asjsm.15590).

English revision needed please.

Author Response

We want to thank the reviewers for reviewing the manuscript. We have made the changes according to the comments of the reviewers in the manuscript and answered the comments point-by-point below here. The changes in the manuscript are colored red. We think the manuscript is now suitable for publication.

Kind regards

Roland van den Tillaar

line 12: I'd suggest to replace "IN the concentric" with "DURING the concentric", it's more appropriate to stress the duration of a contraction regimen. It's something that happens during a time interval and not at a fixed instanto of time. Do the same throughout the paper.

Have now corrected it to DURING throughout the article.

line 14: I don't like the name you gave to the experimental group "eccentric condition", as it sounds like the control gropu did not performed any eccentric contraction. I'd suggest to find a new name and replace it throughout the paper. You may want to use the terminology used by Brandenburg and Docherty (J Strength Cond Res. 2002 Feb;16(1):25-32) where they use "DCER group" for dynamic constant external
resistance training (i.e., the control group, you can keep control group if you want) and the "DAER group" for dynamic accentuated external resistance.

Corrected to DAER condition instead of eccentric condition throughout the article.

line 33: "a classic WEIGHT resistance exercise...and IT is used...". There are, indeed, other typer of resistance so better to specify.

Corrected for and thereby added WEIGHT for more specify.

line 40: "usually" ??? Of course it is: the external resistance is always the same. This "usually" is either redundant or misleading.

We don’t think it is misleading since the bench-press is “usually” performed with the same weight in the concentric and eccentric phase, however, that is not the case for the present study and some previous studies. It in not always the same for this study since the eccentric load is at 110% and concentric load is at 85% for the DAER group.

line 47-49: please introduce to the reader how it is possible that a 100% 1RM load lifted during a bench-press exercise (concetric phase) can result in a 105-120% 1RM during the eccentric phase. This because you use the expression "the same absolute load" which may be misleading. I think the authors are giving too many things for granted here. I think you should introduce first the general concept of dynamic accentuated external resistance (DAER).

Have now described DAER and thereby less confusion for readers.

line 84: 10 participants seem like you've got a small sample. Did you performed any sample size analysis (perhaps based on the previously published literature) so assess whether this sample size is sufficient or not? If you can't support this choice from a  statistical point of view, then discuss it as a limitation.

Yes it is a small sample, however, the sample size analysis was sufficient. We have done previous similar studies in this area and found out that only 7 subjects were necessary for this type of research. You never know exactly how many subjects you need exactly for answering the research question since it is a new study. However, based upon the earlier  studies we think that 10 subjects was enough.   

line 94: what protocol did you use for determining the 1RM (direct method)? Pleas specify the relevant literature.

The protocol was described in the methods and this was the same approach as in many of the previous bench-press studies by Van den Tillaar and Larsen that we think it is not necessary to refer to these studies again..

Figure 1: please increase font size of the text within the figure.

Have now increased the fonts in figure 1.

Statistical analysis is fine.

Discussion section: I appreciate thee fact that authors have discussed their results in the light of the relevant previously published research, but your study has no limitations! Please add some discussion on what could have been improved in your experimental research so readers may be aware of it when replicating your experiment. A glimps of "future work" seems to be contained at line 324-327, you may add your limitations before this part. Furthermore, I think that, for the sake of study reproducibility and training methodology, it is important to stress the fact that such an experimental methodology can be achieved only using draw-wire econders. This because there is an increasing trend in using inertial sensors (accelerometers) for assessing the barbell kinematics both at a research and at a consumer level. It has been shown, infact, that the reliability of accelerometers decreases approaching 90% of the 1RM. This because the acceleration of the barbell would be close to the gravitational acceleration in case of very slow movements and the accelerometer would not be able to sense any variation of velocity. In other words, the accelerometer won't notice any movement of the barbell when liftin 100-120% 1RM (see https://dx.doi.org/%2010.5812/asjsm.15590).

Added a limitation part. However, in this study we used a linear encoder which has been seen very reliable for measuring velocities. We have performed a study in which we compared several linear encoders with some IMU based systems and found that almost all of these systems had similar results, slo at 1-RM. We are still analysing that study , but will publish this in the near future.

English revision needed please.

Reviewer 2 Report

General comment

I read with attention the present study investigating the effects of an eccentric overload (110%RM) in bench press exercise on electromyographic and kinematic activity during the subsequent concentric phase in trained males. Overall, the rationale is well explained and the results are interesting and well discussed. I however have some comments that should be addressed.

Line 32-35: if is possibile, please add a reference in support to this statement

Line 73: replace the “method” from the beginning of the paragraph and insert it in after the “subject” section

Line 83: please consider to provide a table with participants' characteristics

Line 107: Sorry if I missed it, but how many familiarization tests were performed?

Line 317-331: please consider to add a limitation section

Line 332: the “conclusions” section is quite limited, consider to expand it

Author Response

We want to thank the reviewers for reviewing the manuscript. We have made the changes according to the comments of the reviewers in the manuscript and answered the comments point-by-point below here. The changes in the manuscript are colored red. We think the manuscript is now suitable for publication.

Kind regards

Roland van den Tillaar

I read with attention the present study investigating the effects of an eccentric overload (110%RM) in bench press exercise on electromyographic and kinematic activity during the subsequent concentric phase in trained males. Overall, the rationale is well explained and the results are interesting and well discussed. I however have some comments that should be addressed.

Line 32-35: if is possibile, please add a reference in support to this statement

Don’t see the need for a reference as it is common to know.

Line 73: replace the “method” from the beginning of the paragraph and insert it in after the “subject” section

We have deleted methods here, because it is already mentioned above.

Line 83: please consider to provide a table with participants' characteristics

We don’t see the need for a table as the information of the subjects is already mentioned in the text.

Line 107: Sorry if I missed it, but how many familiarization tests were performed?

One familiarization test was performed. This is mentioned in the text.

Line 317-331: please consider to add a limitation section

Have now added a limitation section.

Line 332: the “conclusions” section is quite limited, consider to expand it

It is just a study in which we compared one DEAR condition with a control condition. We can only conclude based upon the findings and not come with more conclusions than we have. So we don’t see the need for expanding the conclusion.

Reviewer 3 Report

The paper is well written and provides an interesting approach to this bench pressing technique.   Might only question would be:  In the Materials & methods section, would it be better to have the Subjects section precede the Methods section?  

Author Response

We want to thank the reviewers for reviewing the manuscript. We have made the changes according to the comments of the reviewers in the manuscript and answered the comments point-by-point below here. The changes in the manuscript are colored red. We think the manuscript is now suitable for publication.

Kind regards

Roland van den Tillaar

The paper is well written and provides an interesting approach to this bench pressing technique.   Might only question would be:  In the Materials & methods section, would it be better to have the Subjects section precede the Methods section?  

Have now corrected it. We have deleted methods and only write the procedure after the subjects.

Round 2

Reviewer 1 Report

Thank you to be amenable to my suggestions. Still, some concerns still remain:

1) please add a reference to your previous work for the direct determination of 1RM

2) you say that 10 participants are enough based on your previous experience? This is not enough for a paper. You cannot reply to a reviewer "trust me, iti is sufficient". You have to justify why you think 10 participants are sufficient. You have two ways: 1) you state what you just told me in the limitation section; 2) you show us your power analysis based on your (or someone else) previosuly published data.

3) I have built inertial sensors with my own hands and have been working with these sensors since 2005 and I'm not going to believe in anyone who tells me that with an accelerometer it is possible to estimate the displacement of a 1RM lift by double numerical integration (assuming that an accelerometer could be even able to detect the motion of the barbell). But if you wish you can stay in your position. Anyway, it is a shame you did not discuss this point I raised.

Author Response

1) please add a reference to your previous work for the direct determination of 1RM

Added reference.

2) you say that 10 participants are enough based on your previous experience? This is not enough for a paper. You cannot reply to a reviewer "trust me, iti is sufficient". You have to justify why you think 10 participants are sufficient. You have two ways: 1) you state what you just told me in the limitation section; 2) you show us your power analysis based on your (or someone else) previosuly published data.

We have included this as a limitation to the study.

3) I have built inertial sensors with my own hands and have been working with these sensors since 2005 and I'm not going to believe in anyone who tells me that with an accelerometer it is possible to estimate the displacement of a 1RM lift by double numerical integration (assuming that an accelerometer could be even able to detect the motion of the barbell). But if you wish you can stay in your position. Anyway, it is a shame you did not discuss this point I raised.

In the present study we focused upon the augmented eccentric overload, measured wit a linear encoder. We have not used an IMU and thereby why should we then discuss results measured with IMUs. We are fully aware of the use of IMU limits the accuracy of distance due to double integration, but as mentioned before we did not use an IMU and thereby for this article it is not a point of discussion or limitation. We hope that the reviewer respects our choice of not discussion IMUs in the present study.